

# Day- and night-time aerosol optical depth implementation in CÆLIS

Ramiro González[1], Carlos Toledano[1], Roberto Román[1], David Fuertes[2], Alberto Berjón[1,3,4],
David Mateos[1], Carmen Guirado-Fuentes[1,3], Cristian Velasco-Merino[1], Juan Carlos Antuña-Sánchez[1],
Abel Calle[1], Victoria E. Cachorro[1], and Ángel M. de Frutos[1]

[1]Group of Atmospheric Optics, University of Valladolid (GOA-UVa), Valladolid, Spain
[2]GRASP-SAS, Remote Sensing Developments, Villeneuve D'Ascq, France
[3]Izaña Atmospheric Research Center, Meteorological State Agency of Spain (AEMET), Izaña, Spain
[4]TRAGSATEC, Madrid, Spain

**Correspondence:** Ramiro González (ramiro@goa.uva.es)

**Abstract.**

The University of Valladolid (UVa, Spain) manages since 2006 a calibration center of the AErosol RObotic NETwork (AERONET). The CÆLIS software tool, developed by UVa, was created to manage the data generated by the AERONET photometers, for calibration, quality control and data processing purposes. This paper exploits the potential of this tool in order to obtain products like the aerosol optical depth (AOD) and Ångström exponent (AE), which are of high interest for atmospheric and climate studies, as well as to enhance the quality control of the instruments and data managed by CÆLIS. The AOD and cloud screening algorithms implemented in CÆLIS, both based on AERONET version 3, are described in detail. The obtained products are compared with the AERONET database. In general, the differences in daytime AOD between CÆLIS and AERONET are far below the expected uncertainty of the instrument, ranging the mean differences between $-1.3 \times 10^{-4}$ at 870 nm and $6.2 \times 10^{-4}$ at 380 nm. The standard deviations of the differences range from $2.8 \times 10^{-4}$ at 675 nm to $8.1 \times 10^{-4}$ at 340 nm. The AOD and AE at night-time calculated by CÆLIS from Moon observations are also presented, showing good continuity between day and night-time for different locations, aerosol loads and moon phase angles. Regarding cloud screening, around 99.9% of the observations classified as cloud-free by CÆLIS are also assumed cloud-free by AERONET; this percentage is similar for the cases considered as cloud-contaminated by both databases. The obtained results point out the capability of CÆLIS as processing system. The AOD algorithm provides the opportunity to use this tool with other instrument types and to retrieve other aerosol products in the future.



# 1   Introduction

Atmospheric aerosol particles contribute to climate forcing by their interactions with radiation and clouds and its impact is still subject to large uncertainty (IPCC, 2014). Aerosol measurements are carried out worldwide in order to reduce these uncertainties, using various techniques: active and passive remote sensing (from ground and space) and in-situ. Sun (and
Moon) photometry is one of the most extended techniques for aerosol remote sensing; the main parameter provided by the photometers is the aerosol optical depth (AOD), i.e. the extinction by the aerosol particles in the entire atmospheric column. AOD is a proxy of the aerosol load in the atmosphere; its variation with wavelength, usually quantified by the Ångström exponent (AE), provides information about the size predominance of these particles (Angström, 1961).

Ground-based photometers use direct Sun (or Moon) spectral irradiance to derive AOD. It is calculated from these mea-
surements using the Beer-Bouguer-Lambert law (Shaw, 1976). The AOD uncertainty depends on the photometer model, but it is usually small, about 0.01-0.02 in daytime. These measurements are therefore considered the "ground truth" for calibration/validation purposes.

Ground-based photometer networks provide long-term and near-real time aerosol data that are used for aerosol property monitoring, satellite and model calibration/validation purposes, and synergy with other instruments. These are the objectives
of the Aerosol Robotic Network (AERONET; Holben et al. 1998), the most extended photometer network, but also similar objectives are pursued by GAW-PFR network (Kazadzis et al., 2018) and SKYNET (Takamura et al., 2004). The aerosol monitoring activity in the photometer networks relies on the standardization of instruments, calibration and processing (Holben et al., 1998; Wehrli, 2000). This is the case for AERONET, in which the standard instrument is the Cimel CE318 photometer. This is an automatic instrument that is able to perform direct Sun observations (and direct Moon in the latest version) as well
as a number of sky radiance scans. Narrow band filters and two detectors (Si and InGaAs) allow spectral measurements in the range 340-1640nm. The extinction measurements (Sun or Moon) are taken every 3-15 minutes, and consist of 3 measurements per spectral channel collected within 1 minute. These 'triplets' are the basic measurement for evaluation of the instrument stability and the identification of cloud contamination.

The calibration needed for AOD evaluation is the extraterrestrial signal of the instrument, which is normally derived using
the Langley plot technique (Shaw, 1983) for reference instruments, or side-by-side comparison for field instruments. The reference instruments are calibrated in high altitude stations like Mauna Loa and Izaña (Toledano et al., 2018). Field instruments are calibrated at inter-calibration sites. In the AERONET network, calibration facilities at GSFC/NASA(Greenbelt, USA); PHOTONS/LOA (Lille, France) and GOA/UVa (Valladolid, Spain) are used for this activity. In them, the instruments are routinely calibrated and maintained to ensure data quality. The facilities at Lille and Valladolid are also part of the Aerosol,
Clouds and Trace Gases Research Infrastructure (ACTRIS, www.actris.eu), a pan-European initiative to provide open and high-quality observations of those atmospheric constituents.

There is a need to evaluate the photometer data in real time and control large amounts of data generated by the network. Hence, in order to help in the management of the AERONET/ACTRIS calibration facility at Valladolid, a software tool called CÆLIS was developed (Fuertes et al., 2017). It provides tools for monitoring the instruments, processing the data in real time



and offering the scientific community a new tool to work with the data. For this purpose, CÆLIS contains a database and a web interface to visualize raw data and meta-data, provides processing of sky radiances and supports the monitoring of the instrument performance. This tool is capable to detect several technical problems with the network instruments by an automatic warning system based on the CÆLIS's meta-data and products, which allows a quick response to detect and solve operation

problems.

In this framework, the calculation of the AOD is important because several checks can be applied to the data to ensure the reliability of the measurements. Moreover, CÆLIS also intends to be a framework to facilitate research activities, being the AOD a key product to the present and future investigations. Therefore, the main objective of this paper is to develop and describe the implementation of the aerosol optical depth and cloud screening algorithms in CÆLIS.

The AOD product must be robust and operational: it must work for any site, instrument configuration, etc., and even with incomplete or damaged raw data files, which it should adequately flag if it is the case. CÆLIS is focused on AERONET and the Cimel photometer simply because that is the framework of our calibration activity. And this is actually the best reference point that we have in order to validate the AOD algorithm. Therefore we will compare the results with those provided by the AERONET version 3 AOD algorithm (Giles et al., 2019), including the cloud screening, which is necessary because AOD can

only be derived when the Sun or Moon are not obstructed by clouds.

We present the general framework for the AOD calculation (section 2) and then the daytime (solar) and night-time (lunar) algorithms are described in detail (sections 3 and 4). The latter includes a novel correction (Román et al., 2020) that considerably improves the quality of the lunar retrievals. The cloud screening is described in section 5 and finally the algorithm results are compared with the AERONET database (section 6).

## 2   General framework for AOD calculation

The calculation of the aerosol optical depth in CÆLIS is intended to provide this parameter for a number of instruments, i.e. the photometers within the AERONET network that are calibrated at the University of Valladolid and routinely provide measurements to CÆLIS system. They constitute an operational network with about 40 active sites that deliver data in near real time. Therefore the algorithm needs to be robust and work in a large variety of circumstances: any site location, different

instrument types, incomplete ancillary information, defective input data, etc.

The algorithm to calculate the aerosol optical depth is composed by two main parts. In the first part the algorithm searches into the database all the meta information about the photometer in the specific date-time: calibration coefficients, location, filters, etc. In the second part, the raw measurement data and the meta information are used to calculate the AOD.

Figure 1 shows the process followed to generate the AOD, and as can be observed, it needs a lot of ancillary information.

All that information is being stored in the CÆLIS database. Each photometer on any particular date-time is linked in one deployment ('installation') with all that ancillary information. In this installation the information related with the beginning and ending dates when the photometer was deployed in one site, as well as the coordinates of the site, are stored. Once the photometer location is known, the next step is to know the specific instrument configuration. In this step all the information





related with the instrument type[1], which could change from one deployment to another, is needed in addition to the information about the interference filters of the photometer: e.g. central and nominal wavelength of the spectral channels and the specific gaseous and water vapor absorption coefficients for those wavelengths.

Then, the algorithm reads the photometer calibration (extraterrestrial signal at mean Earth-Sun distance) and the temperature correction coefficients. This information is provided by the Valladolid calibration facility to the AERONET database and is therefore identical to that used in the AERONET version 3 products. The calibration is then adjusted to the Earth-Sun distance for each observation date-time.

The ancillary information needed for the processing is the local pressure and the column of absorbing gaseous species taken into account: ozone, nitrogen dioxide, carbon dioxide and methane. The detailed description on how this information is obtained by CÆLIS for the specific date-time and location is provided in section 2.3. Three levels are established for these ancillary data: 1) Meteorological data fields; 2) Climatology; 3) Standard atmosphere. This is also the hierarchy for the data usage. Therefore a default value, as provided by a standard atmosphere model (for example pressure), will only be used in case the meteorological data fields or climatology table are not available. This approach is intended to provide the necessary ancillary information in a consistent and operational way across the network, even if some sites could provide more accurate values by co-located measurements.

At this point the first main part of the algorithm flow is concluded. A series of flags have been filled in relation to the obtained meta information, and the algorithm enters in the second part, in which the raw data of direct irradiance are processed. The workflow of the computation is shown in Figure 2.

As already mentioned, CÆLIS stores all the data generated by the photometers that are calibrated at our facility. Therefore, the AOD algorithm only needs to get the raw data from the correct table of the database, and use them to calculate the AOD for each measurement. This procedure must be repeated a number of times, that depend on the instrument type. In digital and triple Cimel photometers, equipped with 10 spectral channels, a total number of 30 measurements are collected in each AOD observation. The 3 measurements per channel acquired over 1 minute (triplets) constitute the basic AOD measurement for each wavelength. Temperature correction is applied to these raw data according to the internal temperature recorded at the sensor head (see section 2.3.2 for details).

The total optical depth (TOD) can be computed using the Beer-Bouguer-Lambert law, as shown in section 3, and then, the contributions of molecules and gaseous absorption for each wavelength are subtracted from the TOD in order to obtain the AOD. The precipitable water vapor column (PWV; section 3.2) is also derived from the photometer measurements using the 940nm channel; this PWV value is used to further correct the AOD at 1020nm and 1640nm channels for (minor) water vapor absorption (Smirnov et al., 2004). The Ångström exponent is also calculated from the retrieved AOD values (section 3.2). Finally, the obtained AOD values with 3 observations per wavelength will be screened for cloud contamination (section 5).

---

[1]Three generations of Cimel photometers are nowadays used in AERONET: analog (starting 1992), digital (starting 2002) and "triple" (starting 2013) instruments (Toledano et al., 2018). Within these three families, several versions were developed: standard, extended, polarized, seaprism, etc. Thus a variety of Cimel instruments is in operation in AERONET.





## 2.1 CÆLIS database structure for AOD

CÆLIS is composed of a relational database, a processing module and a web interface (Fuertes et al., 2017). As indicated above, in this database we can find all the information required to compute the AOD. Thanks to the deployment records ('installations') that exist in the database, we can link, for a specific date, all the physical and logical information about each particular instrument and how it is (or it was) configured. That means we can access to the calibration coefficients of each spectral channel, the raw data, the filter specifications, etc. All this information is stored in different tables of the database.

Similarly, after running the AOD algorithm, all the information generated will be stored in different tables of the database. Specifically two tables are designed to store all the AOD information. One table stores the information that is common for all the spectral channels: date and time, site, solar zenith angle, Earth-Sun distance, pressure, algorithm version, etc. All the information stored in the common table is used to calculate the AOD for each channel. It also stores the derived Ångström exponent and PWV values.The second table stores the specific information of each spectral channel. That means we can find in this table the exact central wavelength of the filter, the calibration coefficient and the temperature correction of the channel, the specific absorption coefficient for gases, and the calculated values for the various components (total, Rayleigh, gaseous absorption and aerosol optical depth). According to the CÆLIS database structure (Fuertes et al., 2017), the AOD is a level 1 product ('direct product'), therefore some redundant information is included in these tables in order to facilitate the data extraction by users.

## 2.2 Computing

Thanks to the processing chain of CÆLIS, the near real time provision of AOD can be achieved. Every raw data file that is received in CÆLIS activates a set of triggers. First of all, the AOD algorithm runs between the first and last measurement included in the data file. Once a first version has been calculated, the system checks whether the AOD has been generated using a pressure value obtained by meteorological analysis (section 2.3.1) or only standard pressure was available. If pressure from meteorological analysis was not yet into the database, the system creates a new task (12 hours later), to reprocess the data until analysis pressure data are available.

Each file received by the system activates the task to calculate the cloud screening. This task runs the cloud screening algorithm for the full day, between 00:00 and 23:59 local time, even if the file does not cover the entire day. Once the AOD and the cloud screening have been calculated, the AOD can be used for further calculations. For instance, a task is triggered to calculate a set of quality control flags, some of them using the calculated AOD as input.

## 2.3 Ancillary data

### 2.3.1 Global Data Assimilation System

NOAA's Air Resource Laboratory runs a series of meteorological analyses and reanalyses; one of this is de Global Data Assimilation System (GDAS, see https://www.ncdc.noaa.gov/data-access/model-data/model-datasets/global-data-assimilation-system-





gdas). The GDAS is run 4 times per day, at 00:00, 06:00, 12:00 and 18:00 UTC. Model output is a grid with 1 degree resolution (360-181 latitude-longitude). This grid contains several meteorological fields at a set of pressure levels.

GDAS data are stored in CÆLIS every 6 hours for the purpose of calculating the local pressure for every site of the network. The pressure at the site elevation is calculated from a set of standard geopotential heights and interpolated in time. When

pressure from GDAS is not available, the algorithm uses a standard pressure calculated with the site elevation, based on the US Standard Atmosphere. A flag indicates if the current AOD value is calculated with a standard pressure or using GDAS pressure.

This strategy to obtain pressure for all network sites is similar to the one followed by AERONET using NCEP/NCAR reanalysis data (Giles et al., 2019). Figure 3a presents the scatter plot between the pressure from CÆLIS and AERONET,

where the range of pressure values spans from 660hPa at Teide site (3570m a.s.l.) up to 1030hPa at sea level sites. More than 180 thousand pressure values used for AOD observations are compared in this plot, showing a high correlation between both databases. The differences between local pressure calculated by CÆLIS and AERONET are in general below 2hPa as shown in Figure 3b.

The use of local pressure data is expected in CÆLIS for the future and will simply add another layer on top of the above-

mentioned hierarchy.

### 2.3.2  Temperature Correction

The Cimel photometers are not stabilized in temperature during operation. In turn, the sensor head is equipped with a temperature sensor that allows correcting the measured signals with respect to a reference temperature of 25°C. The correction is based on a laboratory characterization in a thermal chamber. Whenever a hardware element is changed in the photometer head (filter,

detector, electronic card) a new thermal characterization is run for the instrument. The AERONET procedure for temperature characterization of the Cimel photometers is described in detail in Giles et al. (2019).

The information produced during these characterization, i.e. the temperature correction coefficients for each wavelength above 400nm, is stored in the corresponding table of CÆLIS database. These are extracted by the AOD algorithm to correct raw signals according to the corresponding measurement temperature. The function to correct signal with temperature is quadratic

(i.e. two coefficients per channel). Whenever a characterization is not available for a particular instrument or channel, a default standard correction is applied, as produced by AERONET analysis of historical filters, based on the filter manufacturer or type.

### 2.3.3  Climatology tables

The AOD algorithm needs to account for gaseous absorption at the different wavelengths. Several gaseous species are taken into account: ozone, nitrogen dioxide, carbon dioxide and methane. The column amounts of $CO_2$ and $CH_4$ are considered

constant and a fix value of optical depth scaled to local pressure is used to account for these absorptions in the 1640nm channel (Giles et al., 2019). For $O_3$ and $NO_2$ CÆLIS uses climatology tables produced from satellite observations.

These climatology tables are monthly averages assigned to day $15^{th}$ of each month. The column abundance on other days is obtained by temporal interpolation. The $NO_2$ climatology was obtained from OMI version 3 (OMNO2d (gridded) Level 3,





Krotkov et al., 2017) data between 2005 and 2017. An example of global $NO_2$ for the month of August of this climatology can be seen in figure 4. For the $O_3$ climatology we use Multi Sensor Reanalysis from GOME-2, OMI and SCIAMACHY sensors between 1978 and 2008 (van der A et al., 2010). An example, in this case the global values $O_3$ for the month of May, can be seen in figure 5, where higher ozone values are observed in the Northern hemisphere, as expected in spring.

5     The comparison between the climatology tables used in CÆLIS and AERONET for $NO_2$ and $O_3$ are shown in figure 6 by means of frequency distributions of the differences. For $NO_2$, the determination coefficient between CÆLIS and AERONET is high ($R^2$=0.978), and the mean of all differences (-0.04 DU) points out a little underestimation of CÆLIS database to AERONET climatology values with a standard deviation around 0.02 DU. In the case of $O_3$, the scatter plot indicates very good correlation ($R^2$=0.995); the departure is typically within $\pm5$ DU with a mean bias close to zero and a standard deviation 10   around 2.5 DU.

    For calculation of the absorption optical depth of these species, the spectral absorption coefficients provided by Gueymard (1998) are applied, taking into account the spectral response functions of the individual filters.

## 3   Direct Sun algorithm

### 3.1   Aerosol optical depth

15   The basic equation for aerosol optical depth calculation is the Beer-Bouguer-Lambert law (Shaw, 1976; Cachorro et al., 1987). In practice, this equation is applied to the raw instrument signal at a given wavelength that is measured at ground level ($V$) and the signal that the photometer would have at the top of the atmosphere ($V_0$) (equation 1):

$$V(\lambda) = V_0(\lambda) \cdot R^2 \cdot e^{-\tau(\lambda) \cdot m} \tag{1}$$

    In this equation $R$ is the Earth-Sun distance in astronomical units, $m$ is the optical air mass that indicates the relation between 20   extinction in the vertical column and that in the measurement (slant) path, thus related to the zenith angle of the target (Sun, Moon, star); and $\tau$ is the TOD. The aerosol optical depth can be then derived by subtracting the contribution to extinction by all other atmospheric components: scattering by molecules (Rayleigh scattering) and absorption by gases at a given wavelength.

    The voltage signal ($V$) has a temperature correction following the equation 2. $C_1$ and $C_2$ are the coefficients for the thermal characterization that are stored in the database and $T$ is the temperature given by the sensor head during the measurement.

25   $$V = V'/(1 + C_1(T - 25) + C_2(T - 25)^2) \tag{2}$$

    An absolute calibration (given by $V_0$) is required for AOD retrieval. In order to obtain the top of the atmosphere instrument signal the Langley plot method can be applied (Shaw, 1983; Toledano et al., 2018) or the calibration be transferred from a reference instrument by side-to-side comparison (Holben et al., 1998). This calibration is supposed to be constant over time



except for the Earth-Sun distance variations. A linear interpolation between pre- and pos-deployment calibration factors is applied.

Different air mass factors $m$ are taken into account for the various species; the reason behind is the different vertical distribution of the gases ($O_3$ is mainly stratospheric, $CO_2$ is uniformly mixed, etc.). Hence equation 1 can be rewritten as:

$$V(\lambda) = V_0(\lambda) \cdot R^2 \cdot e^{-[\tau_a(\lambda) \cdot m_a + \tau_R(\lambda) \cdot m_R + \tau_g(\lambda) \cdot m_g]} \qquad (3)$$

where the 'a' subscript stands for 'aerosol', 'R' for 'Rayleigh' and 'g' for 'gases'. Finally, the aerosol optical depth ($\tau_a$) can be directly calculated from equation 3 by:

$$\tau_a(\lambda) = -\frac{1}{m_a} \cdot \left[ ln\left(\frac{V(\lambda)}{V_0(\lambda)R^2}\right) - \tau_R(\lambda) \cdot m_R - \tau_g(\lambda) \cdot m_g \right] \qquad (4)$$

The gaseous absorptions considered in the processing are the $O_3$, $NO_2$, $H_2O$, $CO_2$ and $CH_4$. The airmass for molecular (Rayleigh) scattering $m_R$ is taken from Kasten and Young (1989), whereas the Rayleigh optical depth is taken from Bodhaine et al. (1999) formula and weighted with local pressure. The $O_3$ air mass is taken from Komhyr et al. (1989). For aerosol and $NO_2$ CÆLIS uses $m_R$, and for water vapor ($m_w$) the formulation given by Kasten (1965). The $CO_2$ and $CH_4$ optical depths (1640nm wavelength) are taken as fixed values of 0.0087 and 0.0047 respectively, corrected by local pressure (Giles et al., 2019). The solar zenith angle used in the airmass calculations is computed following Michalsky (1988).

## 3.2 Ångström exponent & Precipitable water vapor

Once the spectral AOD has been calculated, the precipitable water vapor and the Ångström exponent can be calculated. The AE is defined as the negative slope of a linear regression between the logarithm of AOD and the logarithm of the wavelength (in microns) in a defined spectral range. Two AE are calculated: AE(440-870) for AOD between 440nm and 870nm; and AE(380-500) for the range 380nm to 550nm. AE is expected to be different in the different spectral ranges and it depends on the aerosol type (Eck et al., 1999; O'Neill et al., 2001; Vergaz et al., 2005).

The spectral channel that provides the optical depth in the 940nm water vapor absorption band is used to calculate the PWV. In this channel the extinction is produced by aerosol and molecules as well as the water vapor absorption. Therefore, CÆLIS first estimates the AOD at that wavelength as the extrapolation from AOD(870nm) and AOD(675nm) using the Ångström power law in that particular region. Then, CÆLIS follows the methodology described by Schmid et al. (1996), that requires specific characterization of the 940nm filter function of the photometer. This is based on a series of radiative transfer simulations that provide $a$ and $b$ coefficients, unique for each filter, that are used to model water vapor transmittance $T_w$ in the band for the photometer:

$$T_w = exp[-a(m_w u)^b] \qquad (5)$$





Where $u$ is the water vapor abundance and $m_w$ the corresponding airmass. Taking all this into account, $u$ can be finally derived from the photometer signal in the 940nm channel as:

$$u = \frac{1}{m_w} \left( \frac{lnT_w}{a} \right)^{1/b} = \frac{1}{m_w} \left( \frac{-lnV(940) + lnV_0(940) - \tau_R(940) \cdot m_R - \tau_a(940) \cdot m_a}{a} \right)^{1/b} \tag{6}$$

The calibration factor (extraterrestrial signal) for the 940nm channel is also performed during the routine calibrations to-

gether with the aerosol channels.

## 4    Direct Moon algorithm for AOD

The main difference between lunar and solar photometry is that the Moon reflects solar irradiance instead of emitting visible light by itself. This fact makes that extraterrestrial lunar irradiance significantly changes, mainly with the Moon Phase Angle (MPA), even during one single night. Hence, the accurate knowledge of the extraterrestrial lunar irradiance is needed for lunar

photometry purposes. To this end, CÆLIS computes for each observation the extraterrestrial lunar irradiance at several wavelengths following the method of RIMO model ("ROLO Implementation for Moon's Observation"; Barreto et al. (2019)), which is an implementation of the ROLO (RObotic Lunar Observatory) model (Kieffer and Stone, 2005), making use of the SPICE Toolkit (http://naif.jpl.nasa.gov/naif/toolkit.html) (Acton, 1996; Acton et al., 2018). After that, these lunar irradiance values are multiplied by a correction factor proposed by Román et al. (2020), which depends on MPA and wavelength. Following the

Beer-Bouguer-Lambert law, the AOD can be calculated as follows (Barreto et al., 2013):

$$\tau_a(\lambda) = \frac{ln\left[\kappa(\lambda)\right] - ln\left[V(\lambda)/I_0(\lambda)\right] - \tau_g(\lambda) \cdot m_g - \tau_R(\lambda) \cdot m_R}{m_a} \tag{7}$$

where $\kappa$ is the calibration coefficient for an effective $\lambda$-wavelength; $I_0$ is the corrected extraterrestrial lunar irradiance at the same effective wavelength; $V$ is the photometer signal at the channel of the effective $\lambda$-wavelength and $(m)$ values are the optical air masses calculated by the Kasten formula (Kasten and Young, 1989) using the Moon Zenith Angle (MZA) as input.

The AOD can be calculated at night-time using equation 7 if the calibration coefficient $\kappa$ is known. In this work $\kappa(\lambda)$ is calculated by the so called Gain calibration method (Barreto et al., 2016). This method consists of transferring the solar calibration to the lunar channels. The detectors are the same for Sun and Moon direct irradiance measurements in the Cimel; but, in order to reach a higher signal range, the Moon signal is electronically amplified by a gain factor, $G$, with a nominal value of 4096 ($2^{12}$). Taking into account that the only difference between Sun and Moon measurements is in this Gain factor,

the Sun calibration can be transferred to the Moon channels:

$$\kappa(\lambda) = \frac{V_0(\lambda)}{E_0(\lambda)} \cdot G \tag{8}$$

where $V_0(\lambda)$ is the Sun calibration coefficient and $E_0(\lambda)$ the extraterrestrial solar irradiance (Wehrli, 1985), both at the $\lambda$-wavelength. The Gain calibration is simpler and it is not dependent on the RIMO (or other lunar irradiance model) and it only



requires the calibration of the solar channels, which is routinely provided for AERONET instruments. Hence, CÆLIS calculates AOD at night-time using the stored $V_0$ values and the equations 7 and 8. The UV channel of 340 nm is not considered due to the low Moon signal recorded by the photometer at these channels, which implies a low signal-to-noise ratio. More details about the correction applied to the RIMO values and the methodology of AOD calculation can be found in Román et al. (2020).

## 5  Cloud Screening

The global photometer networks like AERONET run hundreds of sites that are equipped with automatic instruments, that measure continuously. The AOD retrieval requires that the Sun is not obstructed by clouds, therefore an automated cloud screening algorithm is required to remove cloud contaminated AOD data, which in general are higher, present higher time variability and show lower spectral dependence than aerosol data. Many algorithms have been published in the literature, in many cases closely tied to the instruments in particular, although many common principles are frequently used: the temporal variability at different time scales, either on the raw signals or the computed AOD, and the analysis of spectral variation (Harrison et al., 1994; Smirnov et al., 2000; Wehrli, 2008; Khatri and Takamura, 2009). Recently, AERONET improved the cloud screening algorithm with several significant changes, including the addition of aureole radiance checks for detection of thin cirrus clouds (Giles et al., 2019).

A cloud screening procedure is therefore needed in CÆLIS. Given the extensive tests with large data sets performed by Giles et al. (2019) and the improvements shown with respect to the previous algorithm, we have tried to reproduce this algorithm for Cimel photometers as a first step for CÆLIS.

The first step of the algorithm is to determine whether the Sun triplet collected is a valid measurement for AOD computation. In this sense, a minimum signal must be achieved in the measurement in order to warranty that photometer is pointing to the Sun (or Moon), i.e. more than 100 counts in the infrared channels (870 and 1020nm). In addition, if any raw signal is lower than the extraterrestrial signal (calibration factor) divided by 1500, which means total optical depth multiplied by airmass about 7, then the corresponding channel is rejected. Moreover, if the variability of the triplet signal (calculated as root mean square over mean) is larger than 16% in any channel, then the full observation is rejected.

The observations that qualify for AOD computation are then checked for AOD variability. Initially all observations are considered 'cloud-free'. They will be flagged as cloudy if the AOD triplet variability (maximum - minimum) is larger than 0.01 (or $0.015 \cdot \tau_a$, whichever is greater) for 675, 870, and 1020 nm channels, simultaneously. If all three channels exceed this threshold, then the measurement is labelled as 'Large triplet'. From this point on, a number of checks are done by the algorithm, that can result in the remaining cloud-free triplets to be flagged as cloudy. The label will indicate which check was activated. The first checks are related to quality control:

– If airmass is larger than 7, then we apply the label 'airmass_range'.

– We check that the Ångström exponent is within the interval [-1, 4]. Otherwise the data are not realistic and we apply the label 'Angstrom_range'.





Then the set of measurement points within a local day (sunrise to sunset) are analyzed together. Whenever new data are received within a certain day, this part of the algorithm will run for all the data available for that day:

- All cloud-free observations of a entire day are labelled as 'potential_measurements' when the number of remaining cloud-free observations is less than 3 in the day, or 10% of the potential measurements attempted by the photometer in 5   that day.

- The temporal variability of AOD at 500 nm is calculated for each pair of consecutive remaining 'cloud-free' observations; the observation with the largest measurement in the pair is assumed as cloud-contaminated and labelled as 'smoothness_criterion' if the difference is larger than 0.01 per minute. This process is iterative and continues until no further data are classified as cloud contaminated by this criterion or the number of data is less than 3 or 10% of the 10   potential measurements, as indicated above.

- The curvature check for aureole radiance is then performed, as described by Giles et al. (2019). This is a novel approach that takes advantage of the Cimel photometer to measure solar aureole radiances, and it is intended to detect thin cirrus clouds. For this purpose, the curvature of the aureole radiance (1020nm) vs. scattering angle is analyzed. If this flag is activated, then the triplet and all other triplets within 30 minutes (or within 2 minutes for Cimel CE318-T instruments) 15   are flagged 'curvature_check'.

- If an observation is distant by more than 1h from any other measurement, then this point is flagged as 'stand_alone'.

- In case the standard deviation ($\sigma$) of AOD at 500 nm of the 'cloud-free' remaining points in the day is larger than 0.015, then the observations that exceed mean $\pm 3\sigma$ in AOD or AE are labeled as '3-sigma'.

A final step is done to recover observations with high spectral dependence, in case AOD (870nm) is larger than 0.5 and the 20   Ångström exponent (675-1020nm) is larger than 1.2. This prevents from removal of very high aerosol loading cases (occasionally with high temporal variability) due to biomass burning smoke and urban pollution (Giles et al., 2019; Smirnov et al., 2000). The label applied in this case is 'restoration' and it is equivalent to 'cloud-free', although very few data in our subset fulfill this conditions.

All flags mentioned above result in the measurement point not to be considered 'cloud_free', and allow us to identify in the 25   database the reason for the rejection. Thus we can query the database for cloud-free data in a certain site and period; but we can also analyze the cloud screened data and discriminate for any specific check.

The full scheme as it has been described, is applied to solar AOD data. For lunar observations, we have maintained the same analysis and thresholds, except for the aureole radiance check, that cannot be performed at night. Further testing is needed to possibly refine the cloud-screening algorithm for night-time.





## 6    Validation of the AOD algorithm

The photometer data that CÆLIS is currently processing for AOD, are all produced by Cimel photometers belonging to AERONET. CÆLIS uses the same raw data and calibration. Moreover, AERONET is a global reference for AOD monitoring and its data are widely used by the scientific community dealing with aerosol, satellite validation and models. Therefore the most logical approach for validation of the CÆLIS AOD implementation is to compare it with the one produced by the AERONET version 3 direct Sun algorithm (Giles et al., 2019). This comparison is provided in section 6.2. The performance of the cloud screening algorithm for this daytime AOD is given in section 6.4. As for the night-time (Lunar) algorithm, section 6.3 includes an analysis of the performance at several sites and Moon phases, but it is not compared to the AERONET processing because the lunar-derived AOD in AERONET is still marked as a provisional product.

### 6.1    Data set for validation

In Table 1 we summarize the data set that has been selected for AOD validation. It comprises two years of data for 9 sites, with about 180 thousand AOD observations (triplets) collected with Cimel photometers.

   The site list includes two high-mountain observatories used for Langley plot calibration of the reference instruments: Izaña and Teide (Toledano et al., 2018). The AOD is very low in these locations, therefore they are very suitable for a detailed comparison. We have also included a rural continental site (Palencia), our calibration site at Valladolid (small city and continental climate), an urban site (Munich), a coastal site (El Arenosillo), a Caribbean site (Camagüey), and the Arctic sites Andenes and Ny-Ålesund. Thus, we have tried to cover several aerosol types and ambient conditions in order to test the robustness of the algorithm.

   Another important aspect of the subset is the variety of Cimel photometer types. We have covered all generations of Cimel instruments (analog, digital and triple) and multiple versions (see Table 1). This feature involved considerable work to ensure a flexible enough algorithm and adequate database construction, so that all data can be consistently processed. In turn, we expect this experience will be of help in the addition of new photometer types to CÆLIS.

   Thus, we have analyzed a large amount of data to have statistical strength in the comparison and cover multiple situations. The data set is used for validation of the daytime (solar) algorithm. This data set will also be used for cloud screening comparison (section 6.4), in which a variety of climate conditions is also crucial.

### 6.2    Daytime AOD validation

The AOD obtained with direct Sun algorithm has been compared for the above-mentioned set of Cimel data. Identical raw data, calibration coefficients and temperature correction factors are used, therefore the differences can only be attributed to the algorithm and the ancillary data sources.

   The criterion for AOD comparison between 2 instruments recommended by the World Meteorological Organization is the so-called U95 threshold (WMO, 2005), defined as:

$$U95 = \pm(0.005 + 0.010/m) \tag{9}$$


Where $m$ is the airmass. As it can be seen in the following analysis, the boundaries of U95 are in general too large for our case, in which we compare algorithms rather than instruments. But it is a good reference as starting point, because it is commonly used in this kind of studies (e.g. Cuevas et al. (2019)).

The AOD comparison for the different wavelengths is shown in Figure 7. The differences are computed as CÆLIS-AERONET
and they are plotted as a function of airmass. The U95 boundary is also depicted, as well as the maximum and minimum difference for each channel. The largest differences are observed for the 340nm and the smallest for the 870nm channel, with $5.1\pm8.2\times10^{-4}$ and $-1.3\pm3.4\times10^{-4}$ respectively. This result is expected because no gaseous corrections are needed in the 870nm channel, therefore the differences can only be caused by different values of the solar zenith angle and the derived airmass. This is an important result, because it indicates that the ancillary data play a key role in the different processing
schemes.

The AOD differences are somewhat site dependent. Apart from the site coordinates (mainly latitude), that conditions the minimum airmass values available for each site, the main relevant difference among sites is the elevation, which affects both the Rayleigh calculations with Bodhaine's formula and the correction by local pressure. The Rayleigh optical depth is larger at the shorter wavelengths, and the analysis of this component indicates that it is the main responsible for the AOD differences
for all channels between 340 and 870nm. The differences in Rayleigh optical depth and AOD clearly decrease for increasing wavelength until 870nm. The differences in pressure for the investigated observations are shown in figure 3b. The mean difference is close to zero, and the standard deviation is 1 hPa. This is noticeable in short wavelengths: at 340nm the Rayleigh optical depth is about 0.70 and 1-2 hPa would mean 0.0007 to 0.0015 optical depth. This fact accounts for half of the discrepancy. The rest can be attributed to the gaseous corrections in this channel ($O_3$ and $NO_2$).

We also noticed an increase in the AOD discrepancy for the longer wavelengths (1020 and especially 1640nm). In this case the Rayleigh correction is minor, therefore we investigated which elements are causing this. As for 1020nm, the water vapor absorption correction is the reason for the slightly worse agreement of the 1020nm channel. The discrepancy is higher for the 1640nm wavelength. The gaseous corrections are in this case responsible for the AOD differences, i.e. the water vapor absorption and the $CO_2$ and $CH_4$ absorption, which are also affected by the differences in pressure.

Overall, the mean of these differences range from $-1.3\times10^{-4}$ at 870 nm to $6.2\times10^{-4}$ at 380 nm. The standard deviation of the differences range from $2.8\times10^{-4}$ at 675 nm to $8.1\times10^{-4}$ at 340 nm. The largest discrepancies are related to the Rayleigh correction (including pressure) as well as the gaseous absorption corrections. The U95 criterion is fulfilled in any case, and most of the spectral AOD observations agree within 0.0015 in AOD, which is one order of magnitude lower that the nominal AOD uncertainty (0.01-0.02) for AERONET field instruments.

**6.3   Night-time AOD evaluation**

The lunar-derived aerosol optical depth has been developed in the last years, after the publication of the ROLO model (Kieffer and Stone, 2005) and the appearance of commercially available lunar photometers (Berkoff et al., 2011; Barreto et al., 2013). It is still a provisional product in AERONET. In this paper we have presented the CÆLIS implementation of the latest improvements in lunar photometry that aims at providing good continuity between solar- and lunar-derived AOD observations.





As it has been shown in previous works (Barreto et al., 2017, 2019) it is important to assume that night-time AOD uncertainty is larger that the uncertainty of daytime retrievals, and that it will also depend on Moon phase angle. These facts also pose additional difficulty to the cloud screening, apart from the lack of the aureole radiance check to detect thin clouds.

The CÆLIS night-time AOD retrievals at several sites and Moon phase angles have been computed, showing the continuity with daytime retrievals. We have intentionally selected cases with low AOD in general, because absolute errors are easier to detect in low AOD scenes. Moreover, we avoided using Izaña site for this comparison, because lunar measurements at Izaña were used to elaborate the correction proposed to improve AOD (Román et al., 2020).

The results for both the AOD and the AE are shown in Figure 8. The upper plot (Andenes site) corresponds to a first quarter case (MPA about -80°). The middle panel (Valladolid) is a full Moon case, in which we can see the step from negative to positive MPA in the full Moon. The lowermost panel (Granada) is a third quarter case. The other two panels (for Teide and El Arenosillo sites) are cases with intermediate (negative and positive) phase angles. Overall, the day-night continuity in AOD is excellent (less than 0.02 in all channels), especially if we bear in mind the AOD natural variability and the nominal uncertainty for daytime AOD of 0.01-0.02, larger for shorter wavelengths (Holben et al., 1998; Giles et al., 2019). The night time AOD has no dependence on Moon zenith angle and the AOD wavelength dependence (typical decrease with wavelength) is basically maintained.

The AE has also been included here because this parameter is very sensitive to AOD errors, especially for low AOD (Cachorro et al., 2008). The good continuity and absence of dependence on zenith angle (Sun or Moon) for the AE is a reliable indicator of data quality. The continuity of this parameter is also excellent (about 0.1 or less in absolute terms) for all cases except maybe Teide, because of the extremely low AOD that amplifies the differences in AE.

Note how instrumental noise is visible in Teide data, where AOD is extremely low, whereas for similar phase angle at El Arenosillo, with higher AOD, such noise is not visible. The plot for El Arenosillo is the only case in which AOD at 500 nm is above 0.1. This can be the case for many field sites worldwide and the agreement among spectral channels and with respect to daytime AOD is a clear indicator that the CÆLIS Moon-derived AOD retrieval and the associated correction (Román et al., 2020) perform as expected.

## 6.4 Cloud screening validation

In this section we have compared the cloud screening performance for daytime AOD data only. The AERONET level 1.5 (cloud-screened) data are used for this analysis. The procedure is very straightforward: we analyze the data with a confusion matrix, in order to determine which data assumed as cloud-free by AERONET cloud screening algorithm, are also flagged as cloud-free by CÆLIS, and viceversa. The other two possibilities, i.e. that one algorithm indicates cloud but not the other one, represent the discrepancy between both procedures.

The confusion matrix $C$ is such that $C_{i,j}$ is equal to the number of observations known to be in group $i$ but predicted to be in group $j$ (Pedregosa et al., 2011). Thus in binary classification, the count of true negatives is $C_{0,0}$, false negatives is $C_{1,0}$, true positives is $C_{1,1}$ and false positives is $C_{0,1}$.





Note that the level 1.0 (unscreened data) in AERONET does not include all the measurements attempted by the photometers, because many observations with too low or too variable raw signals do not qualify for AOD level 1.0 computation (Giles et al., 2019). For those data that passed this first requirement, the AERONET cloud screening will select the cloud free cases and include them in the level 1.5 database [2]. According to the flagging system of our cloud screening algorithm in CÆLIS, we have

5 compared the 'cloud-free' or 'restoration' flags with the AERONET level 1.5 database for all the investigated sites and time periods (Table 1).

The cloud screening comparator links all the photometer observations (full triplet) from CÆLIS with its correspondent in AERONET and stores the output in two different arrays, one for CÆLIS and another one for AERONET. The value of 1 will be stored to each database if the observation is 'cloud-free', and 0 if is not cloud-free. With those two arrays the confusion

matrix has been generated and it can be seen in Figure 9, where we indicate the number of observations and the corresponding relative numbers in percent. In the confusion matrix the first row represents all the values that are cloud-free in AERONET, and the second row is for the cloud contaminated data in AERONET. In the same way, the first column represents the cloud contaminated data in CÆLIS, and the second column includes the cloud-free data in CÆLIS.

More than 250 thousand observations have been analyzed here. The results are clearly satisfactory, with more than 99.8%

agreement in the classification. The number of points outside the main diagonal of the confusion matrix is marginal. The analysis of these few observations indicates cases in which minor differences in AOD and AE caused a certain threshold to be exceeded or not (triplet variability, daily standard deviation, etc.) which also can trigger other cloud screening steps, like the potential measurement criteria. We are therefore very confident that the cloud screening in CÆLIS successfully reproduces the performance of the AERONET version 3 cloud screening.

**7 Conclusions**

The CÆLIS software tool was primarily designed to assist in the management of the calibration facility of Cimel photometers at University of Valladolid, associated to AERONET. It provides access to meta data information to the users and intends to facilitate the daily operation of the photometers on site, with the final aim of improving data quality. CÆLIS already provides to users processing of sky radiances and a set of flags to monitor the instrument performance in real time. The AOD product

now complements this tool. Moreover, the AOD is needed for exploiting remote sensing data with the application of inversion algorithms, like GRASP (Dubovik et al., 2014; Torres et al., 2017). These are the reasons behind the development of an operational aerosol optical depth product as well as the necessary cloud screening algorithm.

The implemented AOD algorithm comprises a number of steps, following formulas and procedures that are well established in the literature. The comparison with AERONET version 3 AOD product shows overall agreement better than 0.0015 optical

depth (one order of magnitude less than the nominal AOD uncertainty), the bias and standard deviations being higher for the UV and the 1640nm channels. In the UV this is caused by different Rayleigh computation and gaseous correction, and needs to be investigated further. Similarly, the discrepancy found for 1640nm channel (slightly higher than that of 1020nm) is caused

[2]The AERONET data still pass another validation step regarding quality control checks, see Giles et al. (2019) for details.





by differences in the gas absorption corrections. Such differences, even if they are low, can be significant in case of high altitude stations or polar sites. The AOD retrieved by CÆLIS from Moon observations has shown continuity between day and night-time for different sites and even for low AOD values and Moon phase angles near to the Moon quarters.

5     The cloud screening schemes in CÆLIS and AERONET agree in the identification of cloud free and cloud contaminated scenes by more than 99.8% of the more than 250 thousand investigated cases. For future investigations, we will need to include a site with predominant biomass burning aerosol since this aerosol type was found to be not sufficiently represented in the subset of data used to compare the cloud screening algorithms.

    This paper has shown the capability of CÆLIS to provide AOD values and products with a similar accuracy than AERONET. The architecture of CÆLIS is such, that can be applied to other instrument types or networks. The next planned step is to be 10  able to assimilate and process other photometers different than the Cimel. In this sense, we will be able to apply a common processing to data originated from different photometer types, each one with its own spectral channels, measurement sequence, etc. Note that AOD retrieval and cloud screening algorithms differ for the different existing networks (AERONET, GAW-PFR, SKYNET). The modular approach has proven to be successful in adding several choices to the data processing or assimilating a variety of ancillary data. This will also help incorporating into the system any future improvements such as new gas absorption 15  coefficients, extraterrestrial spectral irradiance of Sun and Moon, etc. The flagging of data allows extracting in a powerful way a subset of data according to the desired criteria.

    Some of the steps in the cloud screening procedure are actually quality control flags. However a full quality control of the AOD product is not implemented yet in CÆLIS and will need to be developed. This approach is especially important for a robust operation of the algorithm and possible near real time applications.

20  *Data availability.* The used data are available from the authors upon request

*Author contributions.* RG, CT and RR designed and developed the main concepts and ideas behind this work and wrote the paper with input from all authors. They also implemented the cloud screening in CÆLIS. RG, DF, CT and AB implemented the AOD algorithm in CÆLIS. CGF, DM, CVM and JCAS tested the algorithm. AC, VEC and AMdF contributed in the interpretetion of results.

*Competing interests.* The authors declare that they have no conflict of interest.

25  *Acknowledgements.* The authors gratefully thank AERONET and PHOTONS teams for the collaboration and support. The authors thank the Spanish Ministry of Science, Innovation and Universities for the support through the ePOLAAR project (RTI2018-097864-B-I00).



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





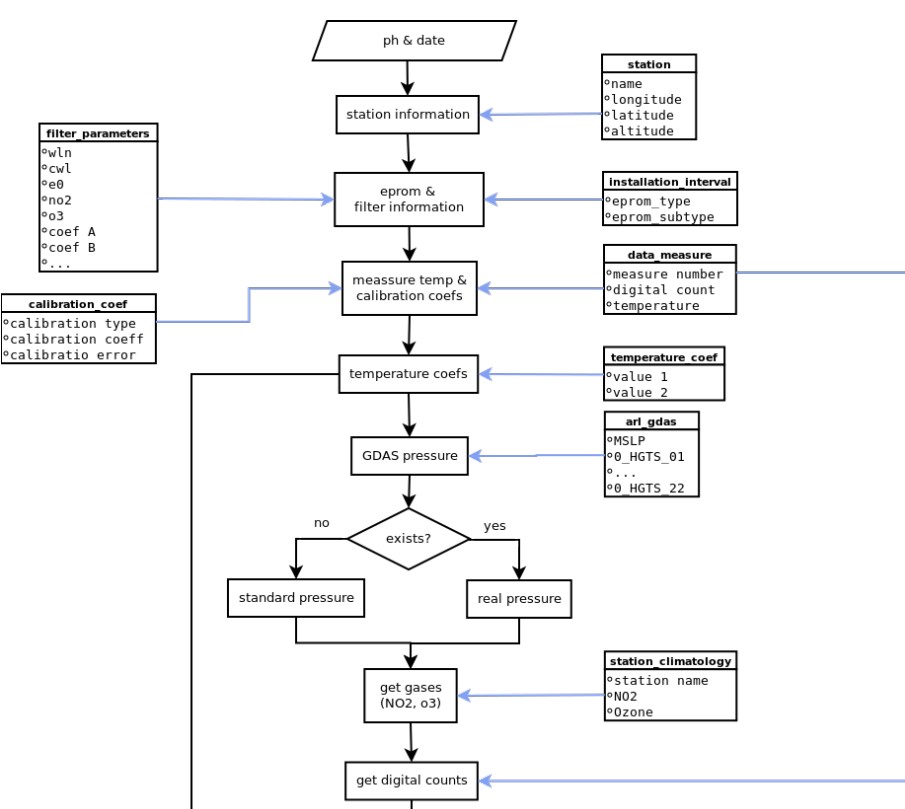

**Figure 1.** Flux diagram of the retrieval of necessary data to be used by the AOD algorithm.





**Figure 2.** Flux diagram of the AOD computation in CÆLIS.





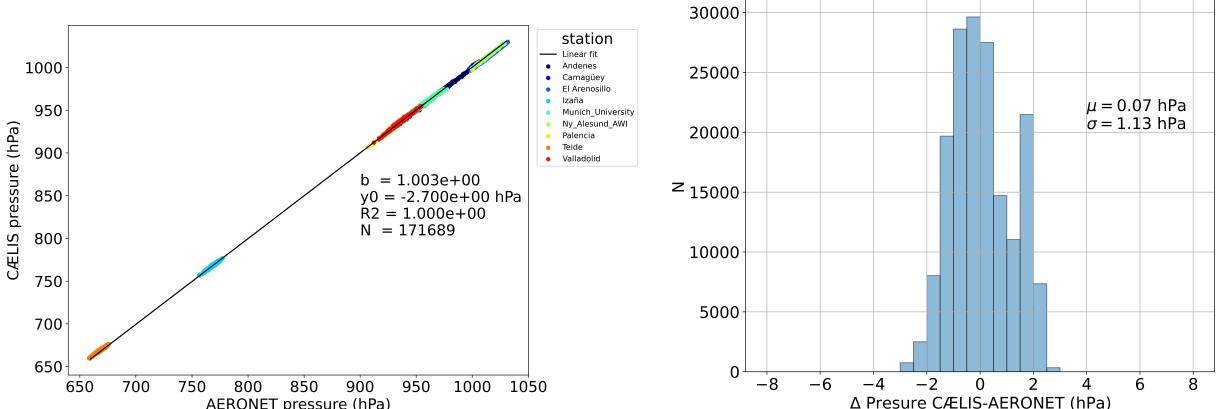

**Figure 3.** (a) CÆLIS atmospheric pressure as a function of AERONET atmospheric pressure for different stations. (b) Frequency histogram of the atmospheric pressure differences between CÆLIS and AERONET databases for all stations.



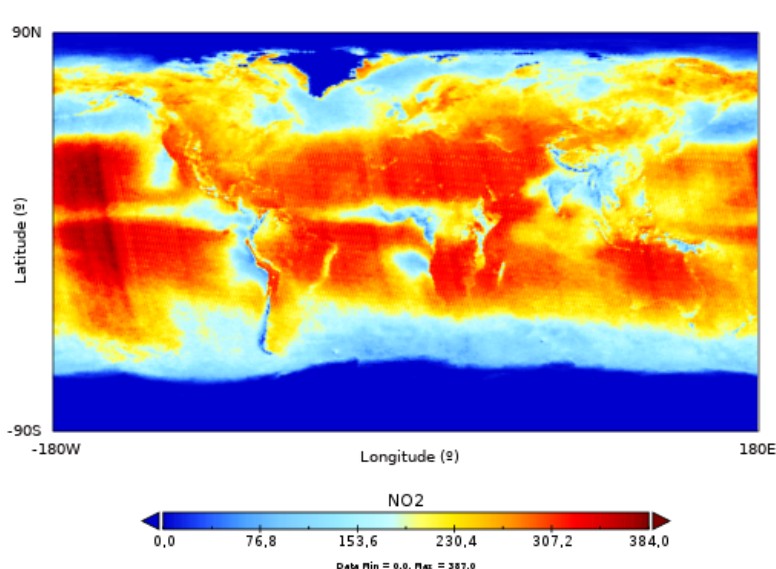

**Figure 4.** Climatology of $NO_2$ (Dobson units $x10^3$) for the month of August. Data obtained from OMI version 3 (OMNO2d (gridded) Level 3) data between 2005 and 2017.



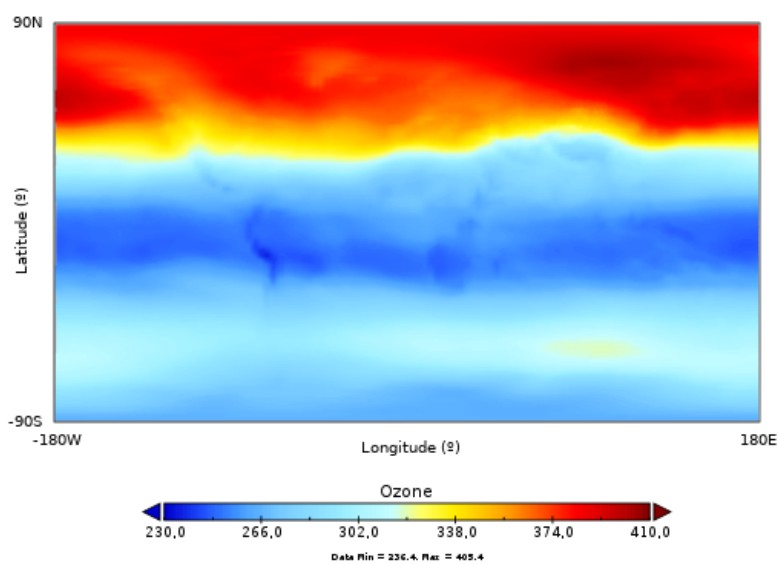

**Figure 5.** Climatology of O$_3$ (Dobson units) for the month of May. Data obtained from Multi Sensor Reanalysis from GOME-2, OMI and SCIAMACHY sensors between 1978 and 2008.





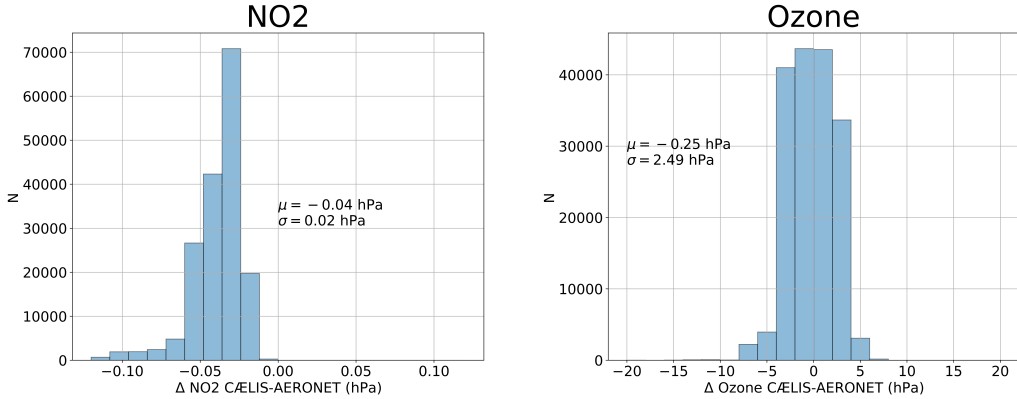

**Figure 6.** Frequency histogram of the differences between CÆLIS and AERONET dabases for: (a) NO₂ climatology; (b) O₃ climatology. Data in Dobson units (DU).



**Figure 7.** Differences in AOD (AERONET-CÆLIS) as a function of airmass for several channels. The red lines indicate the maximum and minimum of the differences. The orange lines indicate the boundaries of the U95 criterion of WMO (2005)

.



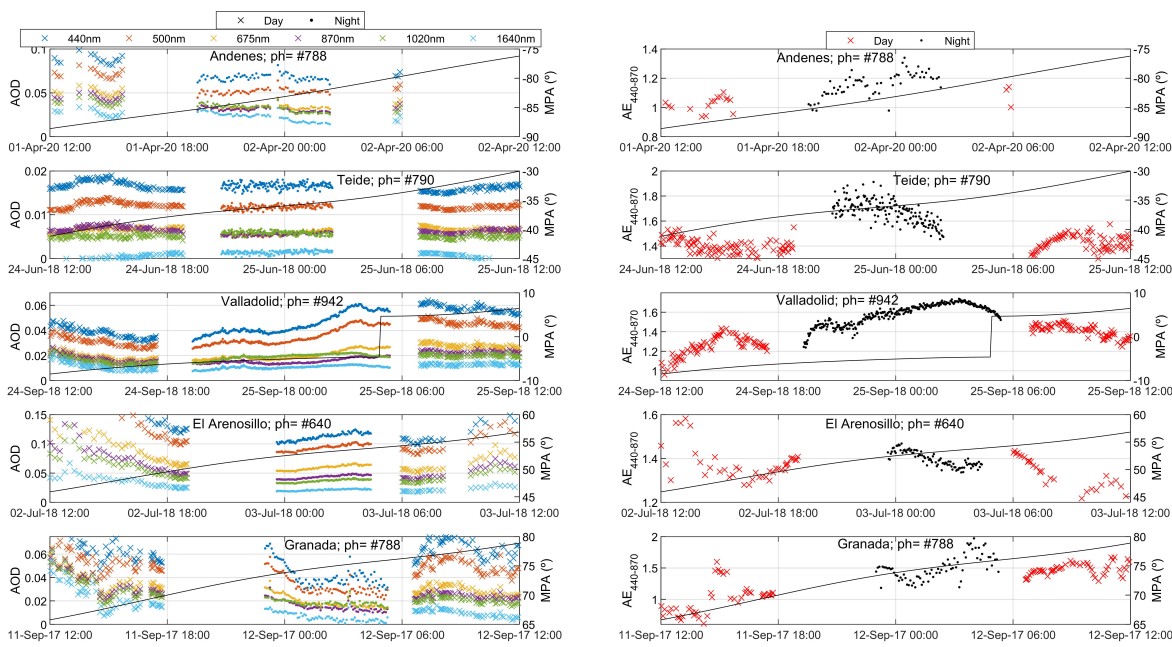

**Figure 8.** (a) Day and nighttime AOD retrievals at different sites and Moon phases. (b) Same for Ångström exponent (440-870nm). The black line indicates the Moon phase angle (MPA, right axis).

.



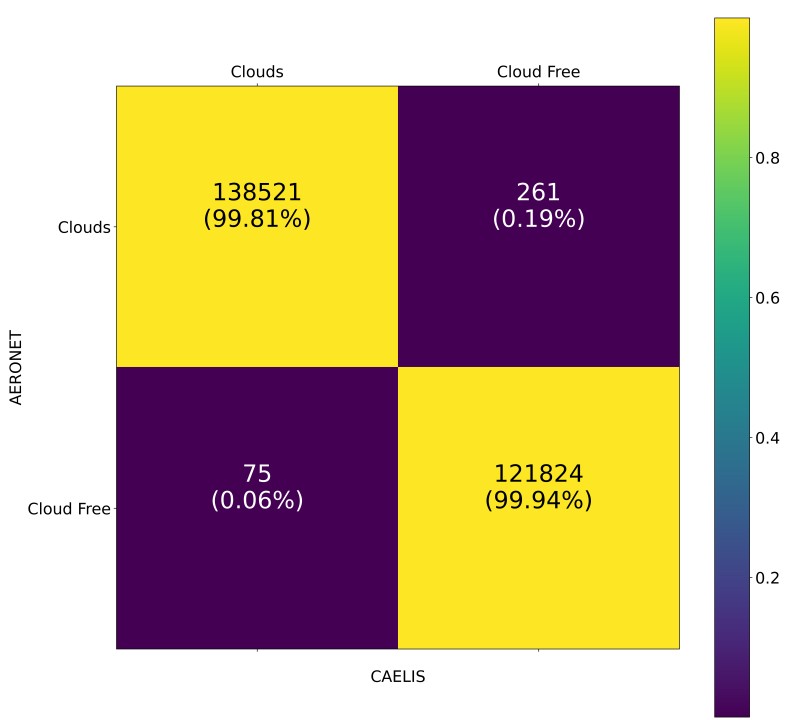

**Figure 9.** Confusion matrix for comparison of the cloud screening performed by AERONET and CÆLIS. Absolute number of cases and relative values (in percent) are given.





**Table 1.** List of Sun photometers used during the validation study. The analyzed period for all sites spans from 2016-Jan-01 to 2017-Dec-31.

| Site | #ph | From | To | Ph type |
|---|---|---|---|---|
| **Andenes** | #904 | 2016-01-01 | 2016-11-20 | Triple - Extended |
| | #789 | 2016-11-21 | 2017-12-31 | Triple - Extended |
| **Camagüey** | #425 | 2016-01-10 | 2016-08-20 | Digital - Extended |
| **El_Arenosillo** | #640 | 2016-05-10 | 2017-04-09 | Triple - Extended |
| | #640 | 2017-07-20 | 2017-12-31 | Triple - Extended |
| **Izaña** | #244 | 2016-01-01 | 2017-12-31 | Digital - Extended |
| **Munich_University** | #198 | 2016-01-01 | 2016-05-17 | Analog - Standard |
| | #600 | 2016-05-18 | 2017-09-06 | Triple - Dual polar |
| | #600 | 2017-11-14 | 2017-12-31 | Triple - Dual polar |
| **Ny_AlesundAWI** | #904 | 2017-06-01 | 2017-12-31 | Triple - Extended |
| **Palencia** | #243 | 2016-01-01 | 2016-10-18 | Analog - Standard |
| | #788 | 2016-10-19 | 2017-03-09 | Triple - Extended |
| | #424 | 2017-03-10 | 2017-05-18 | Digital - Extended |
| | #425 | 2017-05-19 | 2017-07-05 | Digital - Extended |
| | #243 | 2017-07-06 | 2017-11-07 | Analog - Standard |
| | #788 | 2017-11-08 | 2017-12-31 | Triple - Extended |
| **Teide** | #790 | 2016-05-17 | 2016-11-11 | Triple - Extended |
| | #790 | 2017-05-19 | 2017-11-09 | Triple - Extended |
| **Valladolid** | #788 | 2016-01-01 | 2016-05-03 | Digital - Extended |
| | #627 | 2016-05-04 | 2016-10-09 | Digital - Extended |
| | #942 | 2016-10-10 | 2017-03-21 | Triple - Extended |
| | #627 | 2017-03-22 | 2017-07-24 | Digital - Extended |
| | #942 | 2017-07-25 | 2017-12-31 | Triple - Extended |