# Peer review of "Day- and night-time aerosol optical depth implementation in CÆLIS"

_Geoscientific Instrumentation, Methods and Data Systems, 2020_

## Referee Comment (RC1) · Anonymous Referee #1 · 22 Aug 2020

I found this infrastructural contribution very useful for the photometry community, as it is open to CIMEL instruments not fitting all the AERONET requirements, but hopefully also to instrument of different type in the near future.

From my point of view, this can be considered a real research tool, besides being also operational.

It is very well written and clear. I have only two small comments:

- at page 4, lines 8-15. Here you talk about ancillary data such as meteo and gas content. But the text seems to explain only about meteorological parameters, having the 3 options. This is confirmed by the fact that the only option for the gases is the climatology (sect. 2.3.3). Maybe you can adjust the paragraph.

[Figure]

- in Eq 1, the term R2, as defined by you, shouldn't be at the denominator? Please comment.
* * *

---

## Referee Comment (RC2) · Anonymous Referee #2 · 25 Aug 2020

This paper presents a newly developed software tool for analyzing the AERONET data and providing AOD. This tool is robust and can be used on all the AERONET data. The paper clearly describes the software tool and is well organized. The many steps involved are clearly explained as well as the validation procedures.

---

## Referee Comment (RC3) · Anonymous Referee #3 · 1 Sep 2020

Very interesting work on a new algorithm for processing aerosol Data.

A general comment

1 It is not clear what exactly is the purpose of this paper. Is it a comparison of the existing AERONET (NASA based) algorithm with CAELIS ? If this is the case the authors would have to demonstrate what is the added value of using CAELIS versus AERONET/NASA for AOD and Angstrom exponent.

Is it that CAELIS is somehow more accurate ?

Is it that it is an open source code and AERONET / NASA is not ?

When a CIMEL user purchase an instrument what kind of software does CIMEL pro-

vides ?

What algorithm someone has to use in order to be part of the AERONET network ?

So the authors should state more clearly the main reason for introducing CAELIS. The "things" CAELIS could provide in addition to AERONET/NASA and/or the improvements compared to the later.

2 Comparing the two algorithms it is clear that aeronet and caelis use a number of similar inputs, formulas datasets and assumptions. It would be informative to mention in a table or paragraph which are common and which differ.

For example Calibration values, temperature correction functions, filter response use seem common. Pressure, Ozone and NO2 data series are different but these differences are not affecting so much the difference on the datasets. ( here we do not know if they differ from reality).

Cloud detection seem to have some similar aspects but also some new.

So it would be informative for a future Caelis user to know the similarities and differences of the two algorithms

Minor comments

In general this is a comparison of two AOD and Angstrom exponent (AE) processing algorithms: Caelis and the AERONET/NASA algorithm V3 described by Gilles et al.

So in general this have to be more clear in the document. Now you are referring to this NASA algorithm as "AERONET" . You can just write AERONET/NASA algorithm (e.g. ANA) and go through the manuscript with this abbreviation.

Pressure

I think a comparison of caelis and aeronet pressure data could be compared with real pressure data for any sites available in order to have a more pragmatic comparison

Interactive
comment
and to show the impact of the use of these two databases. If this has been already performed in another publication you can just cite it and mention the main result.

Temperature correction.

Why only for wavelengths above 400nm ? What happens to lower wavelengths ?

A linear interpolation on two consecutive calibrations.

Are there any test to understand if a possible change among two consequtive calibration is gradual or a step change ?

You mention "( whichever is greater) for 675, 870, and 1020 nm channels, simultaneously."

I guess due to the decrease of AOD with wavelength if this is a fact for 1020 nm will be for sure a fact also for 675 and 870nm ?

Cloud flagging

Going back to the second comment. It could be informative to state the differences of the two algorithms. The fact that results agree in 99.8% level point to the direction to ask if this can be considered as an improvement or if that the two algorithms agree so it can be just that the main assumptions of the two algorithms are the same.

---

## Author Comment (AC1) · 18 Sep 2020

The authors thank you very much for your comments.

Sincerelly,

Ramiro González
* * *

---

## Author Comment (AC2) · 18 Sep 2020

**Response to the Referee #1 comments for the manuscript "Day- and night-time aerosol optical depth implementation in CÆLIS" By Ramiro González et al. in GID**

**Reviewer comments are in black font (RC), and author comments (AC) in blue font.**

RC: I found this infrastructural contribution very useful for the photometry community, as it is open to CIMEL instruments not fitting all the AERONET requirements, but hopefully also to instrument of different type in the near future.
From my point of view, this can be considered a real research tool, besides being also operational. It is very well written and clear.

RC: I have only two small comments:
- at page 4, lines 8-15. Here you talk about ancillary data such as meteo and gas content. But the text seems to explain only about meteorological parameters, having the 3 options. This is confirmed by the fact that the only option for the gases is the climatology (sect. 2.3.3). Maybe you can adjust the paragraph.
AC: The reviewer is right, we have now separated the meteorological data and gas climatology, because they are treated differently. For those (rare) places where we have no available satellite data for one month, we use the seasonal average, and if the seasonal average is not available we use an annual average to process the AOD.
We have added the following sentence in the manuscript:
"For absorbing gaseous species, we use a monthly climatology (see section 2.3.3). In case some station does not have data for a certain month, a seasonal mean (or annual, if necessary) is used instead. "

RC: in Eq 1, the term R2, as defined by you, shouldn't be at the denominator? Please comment.
AC: Yes, it was a mistake that we have modified it in the revised version. Same mistake also affected equations 3 and 4.

---

## Author Comment (AC3) · 18 Sep 2020

**Response to the Referee #3 comments for the manuscript "Day- and night-time aerosol optical depth implementation in CÆLIS" By Ramiro González et al. in GID**

Reviewer comments are in black font (RC), and author comments (AC) in blue font.

RC:Very interesting work on a new algorithm for processing aerosol Data.
A general comment
1 It is not clear what exactly is the purpose of this paper. Is it a comparison of the existing AERONET (NASA based) algorithm with CAELIS ? If this is the case the authors would have to demonstrate what is the added value of using CAELIS versus AERONET/NASA for AOD and Angstrom exponent.
Is it that CAELIS is somehow more accurate ?
Is it that it is an open source code and AERONET / NASA is not ? When a CIMEL user purchase an instrument what kind of software does CIMEL provides ?
What algorithm someone has to use in order to be part of the AERONET network ?
So the authors should state more clearly the main reason for introducing CAELIS. The "things" CAELIS could provide in addition to AERONET/NASA and/or the improvements compared to the later.

AC: Our purpose is to implement a robust operational AOD algorithm in the CAELIS system, which CAELIS users employ for operational purposes in our calibration facility, field data monitoring, as well as for research.
AOD and sky radiances are the basic quantities provided by the photometers and used for quality control as well as investigations with inversion algorithms, etc. We do not develop a new algorithm (it is all based in well-known literature) nor intend to provide an alternative to AERONET. We compare the results to AERONET because it is robust and well established in the community, and very widely used.
The implementation of such operational code in CAELIS implied a lot of work and we consider it beneficial for the photometry community to have it published. It's not easy to find a detailed description of the algorithms. Especially the small steps or instrument-specific details are often not provided in the literature. And CAELIS users need the description for reference.
This code can be applied to different instruments, as we are currently developing in several projects.
Cimel photometers in particular do not provide any processing software.

RC: 2 Comparing the two algorithms it is clear that aeronet and caelis use a number of similar inputs, formulas datasets and assumptions. It would be informative to mention in a table or paragraph which are common and which differ.
For example Calibration values, temperature correction functions, filter response use seem common. Pressure, Ozone and NO2 data series are different but these differences

are not affecting so much the difference on the datasets. ( here we do not know if they differ from reality).

AC: The aim of the paper is to describe the CAELIS algorithms for AOD. Obviously AERONET is considered a landmark in AOD retrieval, thus we compare our results with AERONET in order to check our results. But we do not want to establish a formal comparison between both algorithms, it would be somehow pretentious and misleading.

RC: Cloud detection seem to have some similar aspects but also some new.
So it would be informative for a future Caelis user to know the similarities and differences of the two algorithms

AC: The cloud screening in CAELIS tries to follow the same steps as in Giles (2019). There are no differences in this sense, although the implementations are different for sure and the small AOD differences also imply in some marginal cases a different output in the cloud screening (see response to last comment).

Minor comments
RC: In general this is a comparison of two AOD and Angstrom exponent (AE) processing algorithms: Caelis and the AERONET/NASA algorithm V3 described by Gilles et al.
So in general this have to be more clear in the document. Now you are referring to this NASA algorithm as "AERONET" . You can just write AERONET/NASA algorithm (e.g. ANA) and go through the manuscript with this abbreviation.

AC: Following the suggestion, we have unified the notation of the AERONET algorithm throughout the text. As mentioned above, the purpose of the paper is to describe the CAELIS algorithm, not to perform an exhaustive comparison with AERONET.

RC: Pressure
I think a comparison of caelis and aeronet pressure data could be compared with real pressure data for any sites available in order to have a more pragmatic comparison and to show the impact of the use of these two databases. If this has been already performed in another publication you can just cite it and mention the main result.

AC: Other authors have already checked the performance of the GDAS pressure product. For instance, Abreu et al (2012) indicate a mean bias of 0.4hPa with standard deviation of 1.2hPa, very similar to our observed differences. This reference has been added to the manuscript.

RC: Temperature correction.
Why only for wavelengths above 400nm ? What happens to lower wavelengths ?

AC: For lower wavelengths, the method used to determine the temperature dependence cannot be used because of low signal. The halogen lamps of the integrating sphere used for this purpose do not have enough flux in the UV range. We performed some tests that indicate very low temperature dependence in general, but statistical significance is poor.

RC: A linear interpolation on two consecutive calibrations.

**Are there any test to understand if a possible change among two consequtive calibration is gradual or a step change ?**

AC: yes, the on-site Langley calibration and also the KCICLO method (Cachorro et al, GRL, 2004) can in some cases, typically clean and clear sites, help to monitor the instrument degradation over time. But this is not done at network level because in many sites it is very difficult.

From our experience, sudden steps occur typically because of dirtiness (spider web, dust, water leakage). Filters degrade slowly (1-2% per year) unless they are too old and start fast degradation. That is the most difficult situation for the temporal interpolation of calibrations.

In case of sudden dirtiness, CAELIS is able to provide a warning and we contact the site manager for cleaning. If change in calibration is larger than 5% per year in some particular channel, we replace the filter in the next maintenance.

**RC: You mention "( whichever is greater) for 675, 870, and 1020 nm channels, simultaneously."**

**I guess due to the decrease of AOD with wavelength if this is a fact for 1020 nm will be for sure a fact also for 675 and 870nm ?**

AC: This sentence refers to the triplet variability, which can be lower for shorter wavelengths, and vice versa, because it depends mainly on moving clouds obstructing the Sun. Note that the measurements are not simultaneously done for all channels.

In order to clarify that, the sentence has been rewritten as follows:

"*They will be flagged as cloudy if the triplet variability (maximum - minimum AOD) is larger than 0.01 (or 0.015\*tau, whichever is greater) for 675, 870, and 1020 nm channels, simultaneously.*"

**RC: Cloud flagging**

**Going back to the second comment. It could be informative to state the differences of the two algorithms. The fact that results agree in 99.8% level point to the direction to ask if this can be considered as an improvement or if that the two algorithms agree so it can be just that the main assumptions of the two algorithms are the same.**

AC: Both algorithms are basically the same, however, the inputs on the algorithm slightly vary due to the very low (but not null) observed differences in the AERONET and CAELIS AOD values (caused by the observed differences on gaseous climatologies and others). These AOD and AE slight differences make in a low number of cases that the output of the algorithm is different. For example, for one observation the CAELIS, the AE value can be 1.001 while the same value is 0.998 for AERONET; in this case if the observation is distant by more than 1h from any other cloud-free measurement, the same algorithm will classify the CAELIS observation as cloud-free while the AERONET one will be classified as stand-alone and removed. In addition, a single point difference between the CAELIS and AERONET output can propagate and produce further differences in cloud-screening; for example, one more cloud-free observation can be crucial to activate or not the "potential_measurements" criterion or to change the 3-sigma threshold.

The stand-alone is better explained in the new manuscript:

"*if an observation is distant by more than 1h from any other cloud-free measurement and it presents an AE (440-870) value below or equal 1, then this point is flagged as 'stand_alone'.*"

The corresponding sentence has been rewritten as follows:

"*In-depth study of these few discrepancies, points out that the differences appear in cases where minor differences in AOD and AE caused a certain threshold to be exceeded or not (triplet variability, daily standard deviation, etc.). Occasionally this also triggered other cloud screening actions, like the potential measurement criteria or 3-sigma threshold.*"